# A machine-learning model for prediction of *Acinetobacter baumannii* hospital acquired infection

Ido Neuman[1]*, Leonid Shvartser[2], Shmuel Teppler[2], Yehoshua Friedman[3], Jacob J. Levine[4], Ilya Kagan[1], Jihad Bishara[5], Shiri Kushinir[6], Pierre Singer[1]

**1** Department of General Intensive Care and Institute for Nutrition Research, Rabin Medical Center, Beilinson Hospital, Petah Tikva, Israel, **2** TSG IT Advanced Systems Ltd., Or Yehuda, Israel, **3** Housetable Ltd., Jerusalem, Israel, **4** Harel Insurance, Jerusalem, Israel, **5** Infectious Diseases Unit, Rabin Medical Center, Beilinson Hospital, Petah-Tikva, Israel, **6** Rabin Medical Center Research Authority, Beilinson Hospital, Petah Tikva, Israel

* idoneuman@gmail.com

## Abstract

**Data Availability Statement:** We are linked to legal restriction since the data are owned by Clalit Health Services, an HMO that is owning our medical institution. The rules of this HMO is only to deliver

### Background

*Acinetobacter baumanni* infection is a leading cause of morbidity and mortality in the Intensive Care Unit (ICU). Early recognition of patients at risk for infection allows early proper treatment and is associated with improved outcomes. This study aimed to construct an innovative Machine Learning (ML) based prediction tool for *Acinetobacter baumanni* infection, among patients in the ICU, and to examine its robustness and predictive power.

### Methods

For model development and internal validation, we used The Medical Information Mart for Intensive Care database (MIMIC) III data from 19,690 consecutive adult patients admitted between 2001 and 2012 at a Boston tertiary center ICU. For external validation, we used a different dataset from Rabin Medical Center (RMC, Israeli tertiary center) ICU, of 1,700 patients admitted between 2017 and 2021. After training on MIMIC cohorts, we adapted the algorithm from MIMIC to RMC and evaluated its discriminating power in terms of Area Under the Receiver Operating Curve (AUROC), sensitivity, specificity, Negative Predictive Value and Positive Predictive Value.

### Results

The prediction model achieved AUROC = 0.624 (95% CI 0.604–0.647). The most significant predictors were (i) physiological parameters of cardio-respiratory function, such as carbon dioxide ($CO_2$) levels and respiratory rate, (ii) metabolic disturbances such as lactate and acidosis (pH) and (iii) past administration of antibiotics.

the data upon request. We have a Chief Innovation Officer, Hagit Hendel, for our hospital who is the manager of data access. She can be reached through this email address - hagithe1@clalit.org.il.

**Funding:** The author(s) received no specific funding for this work.

**Competing interests:** The authors have declared that no competing interests exist.

## Conclusions

Infection with *Acinetobacter baumanni* is more likely to occur in patients with respiratory failure and higher lactate levels, as well as patients who have used larger amounts of antibiotics. The accuracy of Acinetobacter prediction may be enhanced by future studies.

## Introduction

*Acinetobacter baumannii*, a Gram-negative bacterium, possess a major threat upon Intensive Care Units (ICU) worldwide. This virulent bacterium, transmitted by the hospital surroundings, is known to cause mainly bacteremia and nosocomial pneumonia and is associated with increased morbidity, and prolonged hospital stay [1, 2]. The mortality rates for *Acinetobacter baumannii* pneumonia can be as high as 33%, with mortality rates of bacteremia reaching 70% [3].

Risk factors for infection include admission from nursing home residence, immunosuppression history, prolonged ICU stay, invasive mechanical ventilation, enteral feeding, and prior use of broad-spectrum antibiotics [4, 5]. Interventions aimed at containing *Acinetobacter baumanni* outbreaks consist of a combination of surveillance cultures, hand hygiene, careful cleaning of the hospital facilities and surroundings, and contact precautions [2, 6].

Efforts are being made to contain *Acinetobacter* outbreaks and identify patients who require early aggressive antibiotics. One of the primary ways to tackle this issue is by actively screening ICU patients through skin and airway cultures to locate early carriers without any visible clinical disease and examine/quarantine them more thoroughly. Two empirical studies have shown that active surveillance has significant predictive value and reduces the incidence of subsequent infections. Additionally, these studies have demonstrated that early detection may result in reduced mortality and morbidity [6, 7]. Therefore, seeking out other early signs of contamination is a crucial objective.

Machine Learning (ML) models are becoming increasingly popular in healthcare for their ability to capture complex, nonlinear relationships in Electronic Health Records (EHR) data. The Covid-19 crisis has led to a surge in the use of ML-based tools, resulting in the deployment of many decision support systems on the digital interfaces that physicians use every day. These tools enable high acuity surveillance, treatment, and prognostication [8–11].

This study focuses on using machine learning tools to analyze ICU big data and predict clinical Acinetobacter infection. Our primary objective was to create a tool that could help us identify patients who are at a higher risk of getting infected in the near future. Our secondary goal was to determine the key risk factors associated with Acinetobacter contamination, and by doing so, gain a better understanding of the host response to this serious hospital-acquired infection.

## Material and methods

This study is a retrospective cohort study, employing large-scale, comprehensive ICU records.

To create a Machine Learning (ML) model with a higher discrimination power, we chose a two-step approach: (i) first, to train a ML predictive model employing a large external dataset —The Medical Information Mart for Intensive Care III (MIMIC III) (ii) to adapt and test the model using a smaller cohort of Rabin Medical Center (RMC) ICU patients.

## Datasets

MIMIC III comprises detailed anonymous clinical information for more than 52,000 stays in ICUs, at the Beth Israel Deaconess Medical Center in Boston, Massachusetts. These data were collected as part of routine clinical care, between the years 2001 and 2012, and are offered to academic researchers worldwide, courtesy of the hospital [12]. To handle this raw data, we used MIMIC-Extract: A Data Extraction, Pre-processing, and Representation Pipeline for MIMIC III. This open source program provides useful data processing functions such as unit conversion and outlier detection, preserves the time series nature of clinical data, and simplifies MIMIC III database access [13]. To make clinically meaningful predictions, MIMIC extracts aggregate outputs as a sliding window with the size of 6 hour as input features, and then applies a prediction algorithm to predict a clinical event within a 4-hour prediction window, offset by a modifiable (6–24 hours) gap window between the input window and the prediction window (Fig 1).

From RMC hospital, the study utilized data extracted from two types of electronic health records (EHR). The first type was the ICU-specific MetaVision (iMDsoft, Tel-Aviv, Israel) software, which records dynamic sequential parameters such as vital signs, drugs, applied devices, nurse measurements, etc. The second type was the Chameleon (Elad Health, Tel-Aviv, Israel) software, which is a general health record used outside the ICU setting. It compiles demographic data, ER data, general admission times, and data from other hospital sources such as microbiology and laboratory results.

## Study population

**Inclusion criteria.** Patients above age 18; admitted to ICU at BIDC, Boston, US; and RMC, Israel hospitals between the years 2001–2012, and between January 1st 2017-to- February 1st 2021, respectively.

**Exclusion criteria.** ICU stay of less than 48 hours; *Acinetobacter baumanni* culture positive before the time of ICU admission, or during the first 24 hours from admission.

## Research variables

**Primary outcome.** Presence of *Acinetobacter baumanni* infection demonstrated by positive bacterial culture and/or Polymerase Chain Reaction (PCR) test, from any body substance culture (blood, sputum, cerebrospinal fluid (CSF) etc.). A physician reported the time when the culture sample was collected, which was used as the input time for the primary outcome.

**Feature list.** Based on previous studies, clinical experience and feature availability through the EHR, we collected a broad set of variables including demographic features, past medical history list of diagnosis, vital signs, nurse measurements, medications, interventions, lab

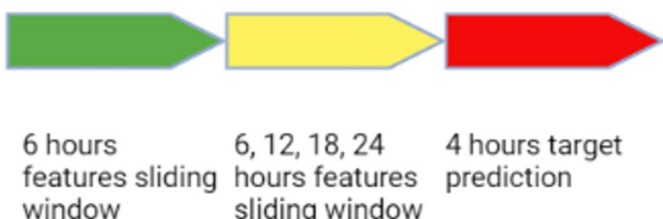

**Fig 1. The sliding window prediction scheme.** The prediction model point of reference is at the end of the 6-hour features sliding window. The model predicted the acquisition of infection at the 4-hour target prediction period, while ignoring the yellow marked gap period.

results, mechanical ventilation (MV) data and ICU scores. A full feature list is presented in S1 Table.

We processed the features by averaging them per hour within 6-hour sliding windows. Missing values were imputed using a modified "Simple Imputation" method, which included a presence mask, the imputed value, and the time since the last observation. Finally, the features were mean-centered and scaled to ensure unit variance.

## ML model development process and statistical methods

**Training.**   Machine Learning models were developed using the MIMIC III database, MIMIC Extract pipeline, Python, PyCharm developer studio, and C++.

During the training phase, the XGBoost algorithm was employed, being recognized for its high speed and accuracy, especially in handling large and medium-size datasets [14]. Our team's extensive experience with different machine learning models and approaches indicated that XGBoost tends to perform particularly well with these types of datasets [11]. The XGBoost model was trained on the MIMIC Extract dataset to predict the likelihood of a patient being infected with Acinetobacter within various time frames, ranging from 2 to 48 hours.

The dataset was randomly split into training (70%), validation (15%), and testing (15%) sets. We utilized 20-fold cross-validation for each time gap. In each of these random folds, XGBoost was initialized with a different random seed to ensure robustness. The mean and confidence interval of the AUC were calculated to evaluate the model performance.

We optimized the model using the Python XGBoost implementation with a grid search approach to modify hyperparameters. The optimal model was selected based on the area under the receiver operating characteristic (AUROC) curve and applied to the testing set for final performance evaluation.

**Adaptation and validation.**   After IRB approval, anonymized raw data was harvested from RMC's databases. The dataset was preprocessed to remove missing values, normalize data, and eliminate redundant features. The researchers accessed, organized, and pre-processed the data between July 2022 and November 2022.

In order to maintain the predictive power of the model trained on the source domain while applied on the target domain, a specialized transfer learning method suited for XGBoost was developed. In the adaptation process, a 20% subset of the target domain's patients were used to fit the source domain model to the target domain, while the remaining patients served as the complementary subset of the target domain for testing. This step enabled translation of Beth-Israel cohort characteristics into RMC standards of patient care. The final step utilized 80% from RMC cohort, as a testing dataset, allowing to examine the final discriminative power of our model.

The discriminative power of the model was assessed using sensitivity, specificity, positive predictive value (PPV), negative predictive value (NPV), and accuracy. We plotted a Receiver Operator Curve (ROC), and calculated the Area Under the Curve (AUC).

Feature importance for classification was determined by XGBoost for the models with 6–24 hour gap in both MIMIC III and RMC data [15]. We chose the model's hyperparameters by random search on a validation set.

The data was presented using mean and standard deviation for normally distributed data and median and interquartile range for non-normally distributed data. Student's t-test and The Mann-Whitney U test were used for variable comparisons. Categorical data was presented as percentages (%) and numbers and compared using Fisher's exact test or $\chi$2 test. To account for the increased risk of Type I errors due to multiple comparisons, we applied the Bonferroni correction to adjust the p-values. The model discrimination was evaluated by the specificity,

sensitivity, positive predictive value, negative predictive value and area under the Receiver Operating Characteristic (AUROC) curve.

**Missing data.** The complexity of the data structure required the application of an imputation algorithm for missing data. The imputation algorithm converted data to a constant rate (1-hour) and produced three features for each measured item: the mean value per hour, a 0 or 1 mask for all measurements within 1-h, and the time since the last measurement, which reflects the measurement's accuracy; the longer the time since the last measurement, the lower the accuracy. Along the fitting procedure, the XGBoost algorithm incorporated these features. It enabled a more trustworthy use of measurements in the face of scant measurements. This technique is commonly employed in ML algorithms for medical data [16].

The study was approved by the IRB (RMC-0392-14) and patient consent was not required because it was retrospective and observational. Data was anonymized by the RMC Research Authority before distribution to the researchers.

## Results

### Study flowchart and descriptive statistics

Following the application of the exclusion criteria, 19,690 MIMIC patients and 226 RMC patients were extracted for our cohort, and used for the empirical analysis. Fig 2 presents the flowchart of the study.

Table 1 presents descriptive statistics of our two samples. MIMIC patients were on average slightly older (62.4 years vs. 59.9 years), heavier (BMI of 28.3 vs. 25.6), and were hospitalized in the intensive care unit for a shorter period (median of 4.29 vs. 18.66 days). Moreover, they

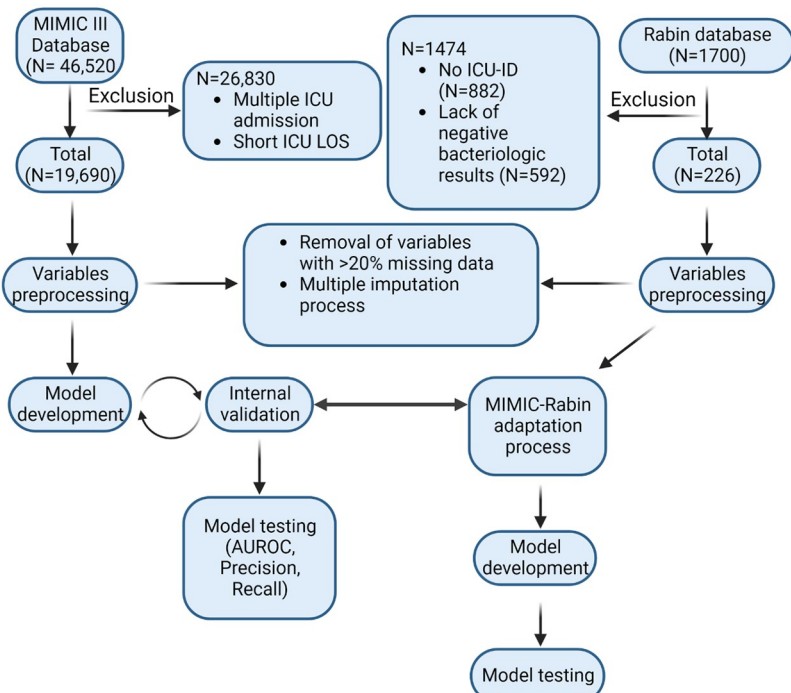

**Fig 2. Study flowchart.** MIMIC: Medical Information Mart for Intensive Care; ICU: Intensive Care Unit; LOS: Length Of Stay; AUROC: Area Under the Receiver Operating Curve.

**Table 1. Descriptive statistics.**

|  | MIMIC III | RMC | P-Value |
|---|---|---|---|
| Total patient number, n | 19690 | 226 |  |
| Mean age [years], (SD) | 62.4 (16.5) | 59.9 (16.1) | 0.023 |
| Male sex, n | 11006 (55.9%) | 154 (68%) | <0.001 |
| Weight] kg], (SD) | 84.8 (25.7) | 77.3 (18.5) | <0.001 |
| Height [cm], (SD) | 168.2 (19.7) | 173.7 (11.7) | <0.001 |
| BMI mean (SD) | 28.31 (1.55) | 25.58 (3.67) | <0.001 |
| Mechanical ventilation, n (%) | 10180 (51.7%) | 172 (76.1%) | <0.001 |
| Acinetobacter rate, n (%) | 214 (1.1%) | 79 (35.5%) | <0.001 |
| HR [BPM] (SD) | 92.1 (18.8) | 90.1 (18.6) | 0.112 |
| MAP [mmHg] (SD) | 78.2 (16.7) | 80.4 (8.4) | 0.048 |
| Creatinine [mg/dl], median (IQR) | 1.2 (1.3) | 1.06 (0.85) | 0.32 |
| Bilirubin [mg/dl], median (IQR) | 0.9 (2.2) | 0.62 (0.93) | 0.27 |
| Platelets [$10^9$/L], mean (SD) | 196 (177) | 185 (119.75) | 0.148 |
| WBC [$10^9$/L], median (IQR) | 8.45 (12.9) | 10.06 (8.86) | 0.12 |
| CRP [mg/dl], median (IQR) | 11.83 (139.35) | 18.5 (25.275) | 0.009 |
| LOS Intensive Care Unit [days], median (IQR) | 4.29 (6.82) | 18.66 (16.05) | <0.001 |
| LOS hospital [days], median (IQR) | 7.98 (9.42) | 21.55 (23) | <0.001 |
| Length of ventilation [hours], median (IQR) | 87 (181) | 333.5 (356) | <0.001 |
| P/F ratio (SD) | 191.97 (93) | 249.7 (142.3) | <0.001 |

BMI: Body Mass Index; HR: Heart rate; MAP: Mean Arterial Pressure; WBC: White Blood Cell; CRP: C Reactive Protein; LOS: Length of stay; P/F: $pO_2/FiO_2$

were less likely to require mechanical ventilation (51.7% vs. 76.1%), and for a much shorter time-period (for those who needed ventilation) (87 vs. 333.5 hours).

Acinetobacter infection was detected in 214 patients in the MIMIC group (1.1%), and 79 patients in the RMC group (35%). It is most likely that RMC's high proportion of infected individuals is due to the fact that we excluded all of the cases that had not been actively examined by culture tests, in order to ensure Acinetobacter negatives are indeed negatives.

## Training model—MIMIC III

The full-feature ML algorithm for MIMIC cohort, utilized 325 features. Among them, only 99 had any added benefit to the model, as the rest of them were weakly predictive (10th percentile or less of the measured weighted feature importance).

Using MIMIC III data, the Acinetobacter prediction model achieved a maximum area under the curve of 0.757 (95% CI 0.72–0.794) after two hours and 0.722 (95% CI 0.675–0.769) after sixteen hours. From the 2-hour gap, the AUC trend degraded, and after 48 hours, the model obtained a plateau of around 0.69 (Fig 3).

Based on a confusion matrix for 6-hours, we see that our model achieved moderate sensitivity and specificity (0.75 and 0.8, respectively) with a high negative predictive value (NPV) of 0.997 (Fig 4).

To develop an explainable ML model, we extracted a feature importance list from the XGBoost tool. The most important contributing factors to the model were advanced hemodynamic parameters (Cardiac Output, Systemic Vascular Resistance), stool out nurse measurements, and several laboratory results (Troponin, Base excess, Creatine Phosphokinase). Fig 5

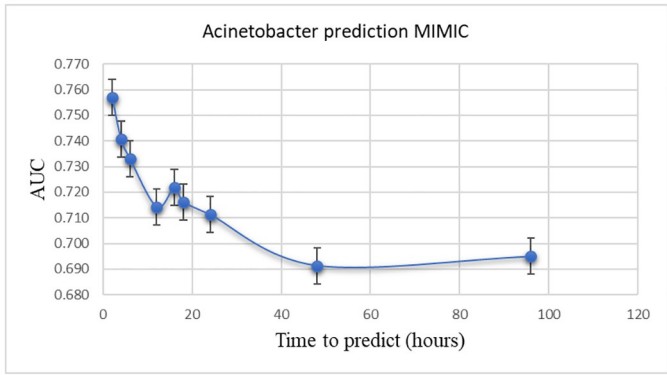

**Fig 3. Acinetobacter prediction MIMIC cohort.** AUC: Area Under the Curve.

presents a list of the 30 most important features, which account for 31.9% of the model's explainability.

## Validation of the model—RMC

Upon receiving an extensive set of 1991 variables from RMC's patients, a comprehensive process of labeling, grouping, and erasing was initiated as described in Fig 6. At the end of the process, 492 features were prepared for use in the prediction model.

Following the MIMIC-RMC adaptation process described above, a final prediction model was developed. Figs 7 and 8 illustrate the prediction ability of the XGBoost model following several different sliding time-window.

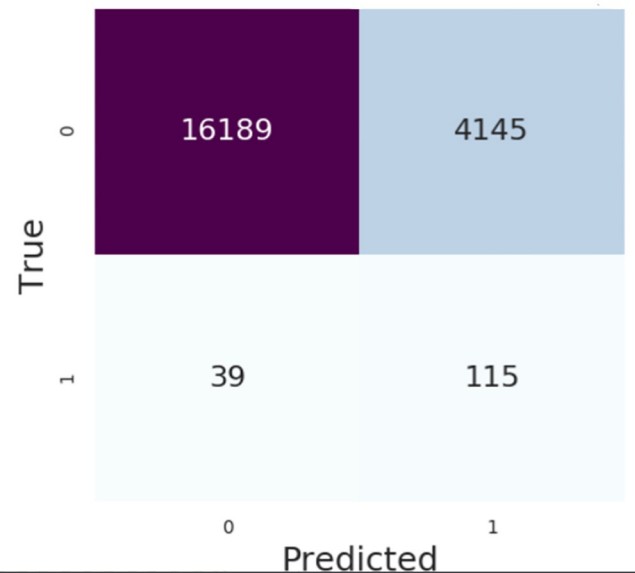

**Fig 4. MIMIC III, 6-hour gap, confusion matrix.** MIMIC III confusion matrix for 19690 patients with a 6-hour gap. Confusion matrix summarizes a binary classifier system's diagnostic capability. Presented numbers are numbers of sliding window intervals used in testing procedure.

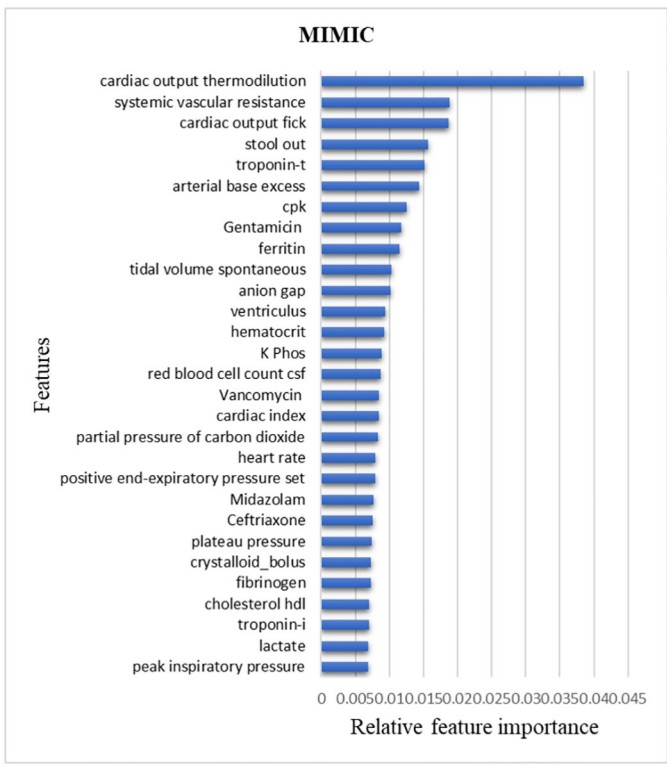

**Fig 5. Feature importance, MIMIC mapping.** The most crucial elements for MIMIC database prediction model are highlighted in relation to one another. In Gradient Boosting Trees (one of which is XGBoost), features are considered relevant based on how much they improve the performance measure, weighted by the number of observations they are responsible for. CPK: Creatine Phosphokinase; CSF: Cerbrospinal Fluid; HDL: High Density Lipoprotein.

The maximal AUC was 0.624 (95% CI 0.604–0.647) when the 6-hour gap was utilized. There was a steady decline in the prediction model with a nadir point occurring at a 24-hour gap window.

As shown in the confusion matrix (Fig 9), the RMC cohort prediction model yielded a sensitivity of 42%, with a specificity of 79%.

Input to the ML algorithm consisted of 492 features, of which only 92 accounted for 90% of the model's elucidability. We found that pH and CO2 lab results accounted for 11% of the explainability of our final model, followed closely by respiratory rate, antibiotic administration, and lactate measurement. Fig 10 depicts 30 of the most influencing features, who all together contributed 65% of the model explainability.

## Discussion

*Acinetobacter baumanni* infections are a serious concern in the ICU, despite isolation and barrier precautions. Guidelines have been provided, but eradication remains a challenge [17].

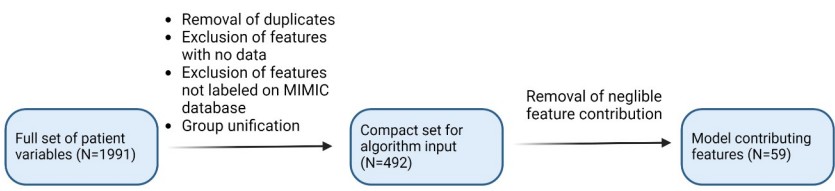

**Fig 6. RMC features preprocessing.**

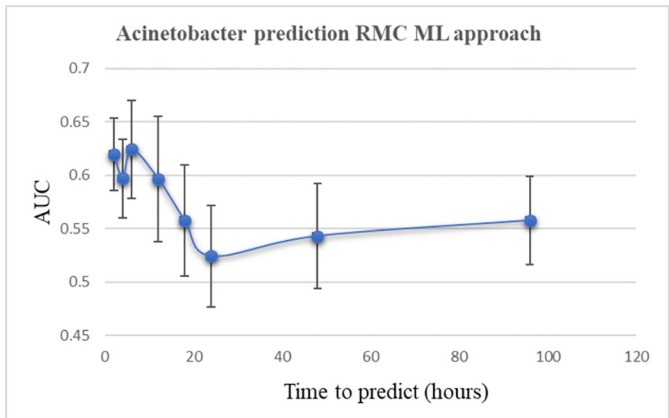

**Fig 7. Acinetobacter prediction, RMC cohort.** AUC: Area Under the Curve.

In this study, we attempted to employ the innovative ML methods and tools, for a careful statistical examination of the factors (demographic, medical, clinical) that affect the probability to get infected with *Acinetobacter*. The use of extensive rich datasets, with numerous features, led to the identification of the elements that affect infection, and also facilitated the examination of the significance and reliability of those elements. As far as we are aware, this is the first attempt to predict *Acinetobacter* hospital acquired infection (HAI) using machine learning.

Our hypothesis was that this type of HAI is predictable. The major advances in Data Sciences, provided us with state-of-the-art algorithms, and thus with the opportunity to arrive at reliable results and conclusions. Employing ML methods on rich datasets will also help with the identification/location of distinct subsets of patients who are more prone to get infected,

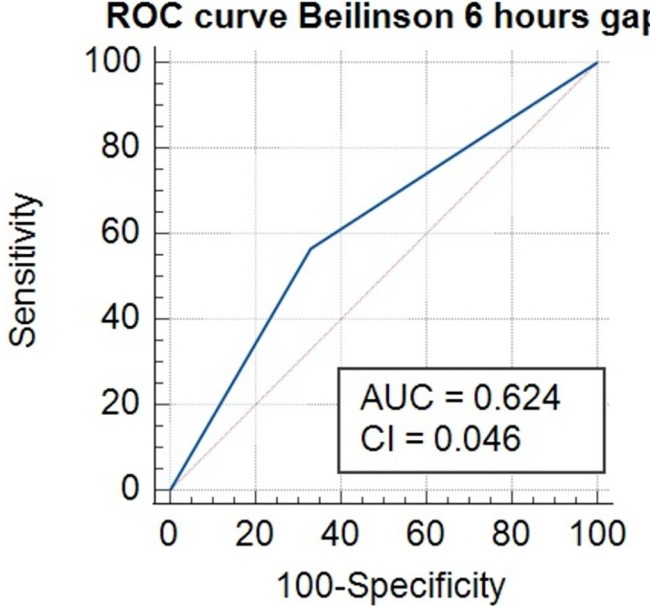

**Fig 8. ROC curve.** ROC: Receiver Operating Characteristic.

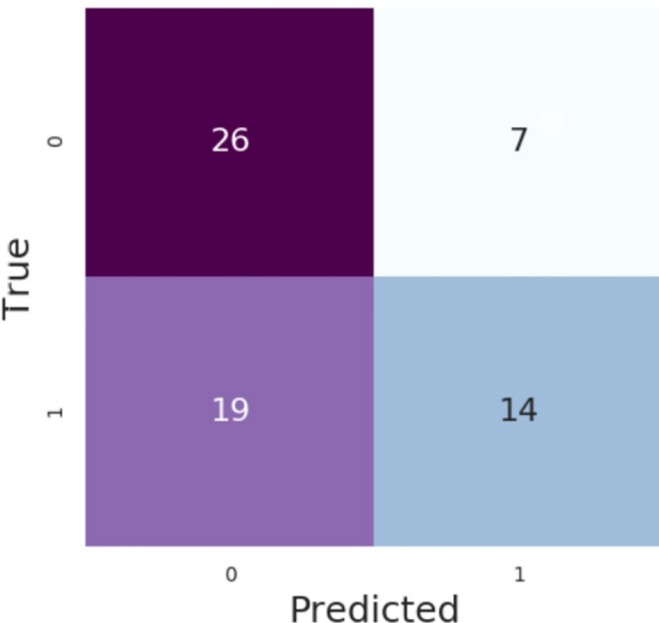

**Fig 9. Confusion matrix, RMC cohort, ML approach.** MIMIC III confusion matrix for 226 patients with a 6-hour gap. Presented numbers are numbers of sliding window intervals used in testing procedure (not patients).

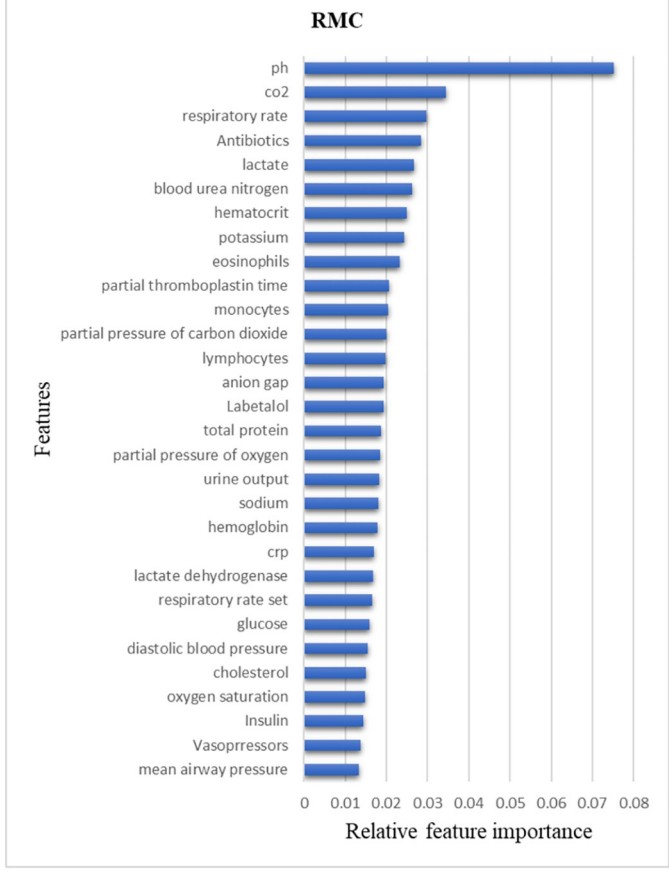

**Fig 10. Feature importance RMC mapping.** CRP: C Reactive Protein.

thus leading to better precautions, preventive measures, and treatment, that will be directed primarily at the more vulnerable patients. Furthermore, identifying high-risk groups could enable us to target future interventions toward a more prone population, where there will be a higher chance of success and a greater statistical power. A deeper understanding of the pathogenesis of healthcare-associated infections (HAI) may also lead to new therapeutic directions. Additionally, incorporating this model into clinical decision support systems can facilitate real-time risk assessment, with predictions spanning 2 to 48 hours, thereby prompting timely reviews, diagnostic testing, and preventive measures.

We employed two datasets that complement each other and enrich the analysis and results: (i) the MIMIC extensive database, with 214 cases of *Acinetobacter* infection (ii) RMC data, that has been adapted to the structure of the MIMIC data, in order to arrive to comparable results.

We managed to develop a prediction model with an AUROC of 0.74 for MIMIC III and of 0.624 for RMC (95% CI 0.604–0.647). For RMC patients, the most important predictors were physiological parameters of cardio-respiratory function, such as carbon dioxide ($CO_2$) levels and breathing rate. Further, metabolic disturbances such as lactate and acidosis (pH) as well as antibiotic treatment were considered important.

These predictors add explanatory power to the model, and are consistent with established risk factors for HAI described in the literature [4, 5, 18, 19].

The fundamental strength of our study stems from the solid detailed dataset, that includes various aspects of patient care. Furthermore, the data used for the analysis was derived directly from the objective monitors used at our ICU, and is therefore consistent and reliable. The study design was complemented and improved by the MIMIC-RMC adaptation process, which allowed us to employ also Boston's EHR for training of the model (overcoming differences in healthcare standards and inclusion years), for robust ML model applicable at different Hospitals.

Our study adds to the ongoing effort to utilize ML in infectious control. It contributes to the existing extensive literature on other prevalent HAIs, such as: *Clostridium difficile* infection (CDI)—Wiens et al. developed a prediction model for CDI infection with an AUC of 0.81 [20]; Bacteremia in immunocompromised patients [21]; Post operative surgical site infections [22]; and Bloodstream infection outcome [23].

However, our model suffers from restrictions and drawbacks.

As evidenced in Table 1, the two datasets are quite different including different patient demographics and different length of ICU stays. In particular, stands out the larger percentage of positive cases in the RMC data (35%, compared to 1.1% in the MIMIC data). The huge difference in the shares of positive cases was unavoidable, as we preferred to exclude from the RMC dataset the many cases that were not examined for culture tests.

Thus, combining them and arriving at comparable conclusions regarding data patterns was quite challenging. We attempted to tackle this obstacle using an adaptation process as described above. However, these statistical and computational efforts provide only a partial solution.

It is important to consider the limitations of the XGBoost algorithms such as. (i) **Interpretability**: The complexity of XGBoost may pose challenges in fully understanding its predictions, which could impact the application of the results in clinical practice. (ii) **Overfitting**: While we applied careful tuning, there is still the possibility that XGBoost could be prone to overfitting, which underscores the importance of ongoing validation to maintain generalizability across different patient populations and time frames [24]. Future research with methodological upgrading, such as more careful consideration of features, a larger dataset, and optimized algorithms, may contribute to the improvement of the model.

Three data limitations can affect the generalization of results: (i) the aforementioned danger of overfitting the model to the investigated patient population and the available data, due to the dual-center approach of the study. Nevertheless, the investigated ICU admissions come from diverse sub-populations, and are representative of the whole range of ICU patients; (ii) The study uses existing data collected over four years by multiple clinicians at the ICUs. Though carefully collecting data by the researcher is preferable, it can reduce reporting errors and inconsistencies; (iii) Culture positive results may not be generalizable due to inconsistencies in the laboratory and testing apparatus used, which may vary between institutions or over time.

Another drawback of our study, is the lack of granularity regarding our primary outcome—positive Acinetobacter results. We didn't divide our outcome to source of origin and/or type of resistance pattern. At clinical practice these factors are very relevant and navigate treatment pathways and patient prognosis [25], but we decided to enhance the study's power instead of further subdividing the data.

Finally, some important features were not included in our datasets: (i) cross-infection data, such as nurse and bed allocations [26] (ii) Patient illness severity scores, such as Charlson Comorbidity Index, and presence of multi-organ failure.

## Conclusions

The prediction model established and trained by our team resulted in reasonable predictive power for up to 24 hours prior to the acquisition of *Acinetobacter* infection, which may aid in the decision to commence barrier measures and early antibiotic treatment. The model was trained and validated on a large patient cohort and depended on a trustworthy dataset. It requires additional external validation by other teams in order to evaluate and enhance its effectiveness across a variety of situations and patient populations.

## Supporting information

**S1 Table. A full feature list.**
(CSV)

## Author Contributions

**Conceptualization:** Ido Neuman, Jihad Bishara, Pierre Singer.

**Data curation:** Shiri Kushinir.

**Formal analysis:** Leonid Shvartser.

**Investigation:** Leonid Shvartser.

**Methodology:** Ido Neuman, Leonid Shvartser, Pierre Singer.

**Project administration:** Ido Neuman, Shmuel Teppler.

**Resources:** Ilya Kagan, Jihad Bishara, Pierre Singer.

**Software:** Leonid Shvartser, Yehoshua Friedman, Jacob J. Levine.

**Supervision:** Pierre Singer.

**Visualization:** Ido Neuman.

**Writing – original draft:** Ido Neuman.

**Writing – review & editing:** Leonid Shvartser, Shmuel Teppler, Ilya Kagan, Jihad Bishara, Pierre Singer.

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
