## [Decision Letter · Decision Letter 0]

28 Jun 2024

PONE-D-24-18204A Machine-Learning model for prediction of Acinetobacter baumannii hospital acquired infectionPLOS ONE

Dear Dr. Neuman,

Thank you for submitting your manuscript to PLOS ONE. After careful consideration, we feel that it has merit but does not fully meet PLOS ONE’s publication criteria as it currently stands. Therefore, we invite you to submit a revised version of the manuscript that addresses the points raised during the review process.

We look forward to receiving your revised manuscript.

Kind regards,

*
**Ali Amanati**
*

**Academic Editor**

*
**PLOS ONE**
*

3. In the online submission form, you indicated that [Results of the study can be obtained from Rabin Medical Center's research authority. Further information can be obtained by contacting the Primary Investigator (I.N).]. 

Additional Editor Comments:

Your manuscript [D-24-18204] has passed the review stage and is ready for ‎revision. ‎

To ensure the Editor and Reviewers can recommend that your revised manuscript be ‎accepted, ‎‎‎please pay careful attention to each comment posted underneath ‎this email (also see files attached by Reviewer#3). This way we ‎can ‎‎avoid future clarifications and revisions, moving swiftly to ‎a decision.‎

Technical points:‎

‎1. Please provide a point-by-point response to the Editor and reviewer's comments

‎2. Please highlight all the amends on your manuscript with a yellow color‎

‎3. Use line numbering and page number in the next submission‎

‎

The editor main concerns:‎

‎-Why the “indwelling catheter use” has been not considered? ‎

‎-What justifies focusing on a small number of antibiotics in comparison to the ‎‎‎“overall history of antibiotic use”? ‎

‎-Why the critically ill patient scoring systems like the “Carlson Comorbidity ‎‎Index” have been not considered?

-Why "multi-organ failure" has been not considered (kidney/liver)? ‎ ‎

‎

-Why the resistance pattern (ESBL/Carbapenemase) of Acinetobacter infections ‎has been not considered between two dataset?‎

‎-Taking the infection's origin (source) into account is crucial when dealing with ‎Gram-negative infections. This leads to the classification of certain infections as ‎‎"high inoculum" and others as "low inoculum," and thus, the severity of the ‎influencing factors varies. Researchers have been ignored this point.‎

**Reviewers' comments:**

Reviewer's Responses to Questions

**Comments to the Author**

1. Is the manuscript technically sound, and do the data support the conclusions?

Reviewer #1: Partly

Reviewer #2: Yes

Reviewer #3: Partly

2. Has the statistical analysis been performed appropriately and rigorously? 

Reviewer #1: No

Reviewer #2: No

Reviewer #3: I Don't Know

3. Have the authors made all data underlying the findings in their manuscript fully available?

Reviewer #1: No

Reviewer #2: Yes

Reviewer #3: No

4. Is the manuscript presented in an intelligible fashion and written in standard English?

Reviewer #1: Yes

Reviewer #2: Yes

Reviewer #3: Yes

5. Review Comments to the Author

Reviewer #1: The efforts of the authors are commendable. However, the current manuscript lacks sufficient scientific coherence. The figures presented do not meet the required quality standards, and providing predictive power indicators would improve the output for the reader. Additionally, it is recommended to explore several machine learning methods, comparing and selecting the best model to increase the scientific power of the research. The determination of important features should also be explicitly detailed. If K-fold cross-validation techniques were employed, this should be clearly stated. The discussion could be strengthened. It is advised that the manuscript be revised and rewritten, then resubmitted to the journal for further consideration.

Reviewer #2: This is a fascinating study that could play an important role in predicting the impact of Acinetobacter baumannii in healthcare settings, from infection prevention and antimicrobial stewardship to resource management and optimizing patient care. I have concerns about the presented model that the authors need to answer.

1. You have used the MIMIC III dataset to train an ML model and the RMC dataset to test the model. For this, the distribution of the two data must be the same and similar, because the accuracy of the model may decrease or the output may be irrational.

2. Acinetobacter infection rate is 1.1% in MIMIC group and 35% in RMC group. It seems that the distribution of the two data sets is not the same. This can be one of the reasons for the low area under the curve (0.62). Authors have to definitely check the distribution of 2 dataset.

3. The centralization is an important issue in machine learning that is not mentioned in the article. Have the authors centralized the features?

4. The authors should have compared their proposed model with other models such as random forests or support vector machines or even unsupervised models to better display the performance of the introduced model. Was there a reason this comparison was not made?

Reviewer #3: Please see comments attached.

Please see comments attached.

Please see comments attached.

Please see comments attached.

Please see comments attached.

Please see comments attached.

Please see comments attached.

6. PLOS authors have the option to publish the peer review history of their article (what does this mean?). If published, this will include your full peer review and any attached files.

Reviewer #1: No

Reviewer #2: No

Reviewer #3: No

---

## [Author Response · Author response to Decision Letter 0]

30 Aug 2024

Dear Editor and Reviewers,

Thank you for your constructive and helpful review of our manuscript "A Machine-Learning Model for Prediction of Acinetobacter Baumannii Hospital Acquired Infection”. Your feedback, insights, and valuable suggestions - contributed to a significant improvement of our paper. 

In the letter below, we address in detail, point-by-point, each of the comments, and outline the revisions we have made to accommodate your questions/remarks. Our responses are sub-divided into those related to ‘Journal requirements’, Editor’s comments, and then - responses to the comments of each of the three reviewers.

We hope that you find the revision satisfactory, and thank you again for your input.

Sincerely,

Ido Neuman

**References of page numbers and lines correspond to the clean version of the manuscript**

Style has been revised (where needed), in order to fully meet the style requirements of PLOS ONE.

The code to run this procedure and to evaluate it is available at the following GitHub repository: https://github.com/lshvartser1959/Acinetobacter

3. In the online submission form, you indicated that [Results of the study can be obtained from Rabin Medical Center's research authority. Further information can be obtained by contacting the Primary Investigator (I.N).]. 

We appreciate the reviewers' request for data availability to ensure the reproducibility and transparency of our data-based research. Our study utilizes two distinct data sources: the MIMIC-III database and data from our local hospital. 

The MIMIC-III database is an open-access resource available to qualified researchers who complete the necessary training and data use agreements, as outlined on the MIMIC-III website.

However, the data collected from the local hospital are subject to strict privacy regulations and Institutional Review Board (IRB) policies. Our IRB committee mandates that this data cannot be shared publicly due to patient privacy concerns. Nonetheless, we are committed to supporting the research community and can provide access to this data on a case-by-case basis following special requests. Researchers interested in accessing our hospital data can contact us directly to discuss the specific requirements and processes involved in obtaining the necessary permissions. We hope this approach balances the need for data transparency with the ethical obligation to protect patient privacy.

Additional Editor Comments:

Your manuscript [D-24-18204] has passed the review stage and is ready for ‎revision. ‎

To ensure the Editor and Reviewers can recommend that your revised manuscript be ‎accepted, ‎‎‎please pay careful attention to each comment posted underneath ‎this email (also see files attached by Reviewer#3). This way we ‎can ‎‎avoid future clarifications and revisions, moving swiftly to ‎a decision.‎

Technical points:‎

‎1. Please provide a point-by-point response to the Editor and reviewer's comments

‎2. Please highlight all the amends on your manuscript with a yellow color‎

‎3. Use line numbering and page number in the next submission‎

The editor main concerns:‎

‎-Why the “indwelling catheter use” has been not considered? ‎

Please note that we have considered the use of indwelling catheters, by including specific subcategory variables. For instance, "Urine Catheter" (see row 296 in the supplementary file S1) and "Central Venous Line" (see row 62 in S1) are both accounted for in our analysis. These subcategories provide a detailed examination of the different types of indwelling catheters, ensuring a comprehensive evaluation of their potential impact.

‎-What justifies focusing on a small number of antibiotics in comparison to the ‎‎‎“overall history of antibiotic use”? ‎ 

At the data processing stage of the MIMIC dataset, we initially analysed individual antibiotics. However, to streamline our analysis and enhance interpretability, we later grouped these into a broader 'prior antibiotics usage' category (see Figure 10). This approach allowed us to identify and present the most significant results, focusing on the antibiotics that had the most substantial impact on the outcomes of interest. This methodology ensures a clear and meaningful interpretation of the data, highlighting the most relevant findings.

‎-Why the critically ill patient scoring systems like the “Carlson Comorbidity ‎‎Index” have been not considered?

-Why "multi-organ failure" has been not considered (kidney/liver)? ‎ ‎

These are indeed important and valid points. Unfortunately, our dataset did not include the level of granularity that is required in order to incorporate scoring systems for critically ill patients. We were indeed aware of this limitation, and have acknowledged it in the paper (see page 18 rows 320-322)

-Why the resistance pattern (ESBL/Carbapenemase) of Acinetobacter infections ‎has been not considered between two dataset?‎

‎-Taking the infection's origin (source) into account is crucial when dealing with ‎Gram-negative infections. This leads to the classification of certain infections as ‎‎"high inoculum" and others as "low inoculum," and thus, the severity of the ‎influencing factors varies. Researchers have been ignored this point.‎

As stated in our primary outcome description (see page 7 rows 101-104) we employed a broad definition of positive culture tests, encompassing any source of origin (excluding skin/mouth/rectal screening) and any resistance pattern. The objective of this inclusive approach was to capture a wider range of data and a larger sample. As a result, crucial sub-classifications, such as ESBL and Carbapenemase resistance patterns, are missing, as well as distinctions between "high inoculum" and "low inoculum" infections.

We were therefore faced with the trade-off between a larger sample (that leads to more reliable/significant results) versus the inclusion of sub-classifications. Our decision to favor a larger sample at the cost of ignoring sub-classifications was driven by the limited number of positive results in our datasets. We were concerned that adding subgroups would weaken the statistical power of our results. Being fully aware of the importance of these missing factors, we have acknowledged this limitation in our manuscript (Page 18 rows 315-319). Future studies, with larger datasets, should indeed include these detailed sub-classifications in order to provide a rigorous understanding of the determining factors in Gram-negative infections.

.

5. Review Comments to the Author

Reviewer #1: The efforts of the authors are commendable. However, the current manuscript lacks sufficient scientific coherence. The figures presented do not meet the required quality standards, and providing predictive power indicators would improve the output for the reader. Additionally, it is recommended to explore several machine learning methods, comparing and selecting the best model to increase the scientific power of the research. The determination of important features should also be explicitly detailed. If K-fold cross-validation techniques were employed, this should be clearly stated. The discussion could be strengthened. It is advised that the manuscript be revised and rewritten, then resubmitted to the journal for further consideration.

We provided predictive power indicators in Figures 3, 7, 8 and in the text (see page 13 rows 192-200 and page 15 rows 232-235). 

The reason for using XGBoost method, is based on the experience and findings of our past research, which indicated that XGBoost outperforms the other methods including logistic regression, random forest, convolutional neural network and long short-term memory. We detailed our results in a recent paper (Bendavid, Itai, et al. "A novel machine learning model to predict respiratory failure and invasive mechanical ventilation in critically ill patients suffering from COVID-19." Scientific Reports 12.1 (2022): 10573.). In order to clarify this point, we added a paragraph in the methods section, explaining why we chose XGBoost (page 8 rows 116-121). 

We utilized 20-fold cross-validation for each time gap. In each of these random folds, XGBoost was initialized with a different random seed to ensure robustness. The mean and confidence interval of the AUC were calculated to evaluate the model performance. Additionally, feature importance was determined using the built-in feature importance functionality of XGBoost, which ranks features based on their contribution to the model's decision-making process. To clarify this point, we added a relevant paragraph at page 8, rows 122-125. 

As noted above (in the letter to the reviewers), the manuscript has been extensively revised, and relevant parts have been rewritten. 

‏Reviewer #2: This is a fascinating study that could play an important role in predicting the impact of Acinetobacter baumannii in healthcare settings, from infection prevention and antimicrobial stewardship to resource management and optimizing patient care. I have concerns about the presented model that the authors need to answer.

1. You have used the MIMIC III dataset to train an ML model and the RMC dataset to test the model. For this, the distribution of the two data must be the same and similar, because the accuracy of the model may decrease or the output may be irrational. 

We are well aware that the MIMIC III and RMC datasets were not similar. To address this issue, we implemented an adaptation process, as described in the manuscript, by extracting a subset of MIMIC cases to better align with the RMC dataset. Despite these attempts, differences between the datasets may still impact the model's accuracy and outputs. This drawback of our study is clearly noted in our paper (page 17 Rows 292-299)

2. Acinetobacter infection rate is 1.1% in MIMIC group and 35% in RMC group. It seems that the distribution of the two data sets is not the same. This can be one of the reasons for the low area under the curve (0.62). Authors have to definitely check the distribution of 2 dataset. 

We have double-checked carefully the distributions of the two datasets and confirmed that the data are accurate. We are fully aware of the significant difference in Acinetobacter infection rates between the MIMIC (1.1%) and RMC (35%) groups, and have discussed in the paper potential reasons for the higher infection rate observed in the RMC group (see page 11 rows 179-182 and page 17 rows 292-299). This disparity is considered and discussed in the statistical analysis, in particular concerning the lower area under the curve (0.62). 

3. The centralization is an important issue in machine learning that is not mentioned in the article. Have the authors centralized the features? 

Yes, we performed specific transformations to enhance the training process. Every numerical feature was scaled by subtracting the minimum value and then normalized by the range (max - min). Additionally, the feature "Time since last measurement" was centered by subtracting the mean and then normalized by dividing by the standard deviation. These steps ensured that the data were appropriately scaled and centered for optimal model performance (see added paragraph at page 8 rows 109-112)

4. The authors should have compared their proposed model with other models such as random forests or support vector machines or even unsupervised models to better display the performance of the introduced model. Was there a reason this comparison was not made? 

The reason for using XGBoost method, is based on the experience and findings of our past research, which indicated that XGBoost outperforms the other methods including logistic regression, random forest, convolutional neural network and long short-term memory. We detailed our results in a recent paper (Bendavid, Itai, et al. "A novel machine learning model to predict respiratory failure and invasive mechanical ventilation in critically ill patients suffering from COVID-19." Scientific Reports 12.1 (2022): 10573.). In order to clarify this point, we added a paragraph in the methods section, explaining why we chose XGBoost (page 8, rows 116-121). 

Reviewer #3 

The authors developed a model for identifying patients at risk of acinetobacter baumanni infections in the intensive care unit (ICU), a leading cause of mortality in the ICU. An XGBoost model was trained on a novel data set to accomplish this task.

While this work has merit, updates are needed to improve its clarity and to contextualize the

Model’s results. Please find my comments and suggestions below.

Major Comments

1. Study populations (line 99): Are patients with positive cultures in the first 24 hours excluded because they were likely infected prior to their ICU visit? This exclusion criteria should be motivated.

Indeed. We excluded patients with positive cultures in the first 24 hours after admission at the ICU, in order to focus on infections acquired during the ICU stay rather than those already present upon admission. This criterion ensures that our study addresses primarily nosocomial infections rather than infections present on admission. Thank you for bringing up this point. It led to the important clarification now added to the paper (page 7 rows 98-99)

2. Study populations: Further discussion on systematic differences between the BIDC and RMC populations is needed to contextualize the predictive algorithm’s results. In particular, why are patients without culture tests excluded from the RMC data but not the BIDC data?

These details may further support the transfer learning method described in the “Adaptation and validation” subsection.

We have expanded on this topic in our discussion (see page 17 rows 292-299), acknowledging the significant differences between the BIDC and RMC datasets. Regarding the exclusion of patients without culture tests, we can confirm that this criterion was applied to the RMC data to ensure a higher confidence level in the training phase of our predictive model. Unfortunately, we do not have specific information on the exclusion criteria used by BIDC. Our decision was guided by the aim to minimize false negatives in the training dataset, which may have contributed to the higher incidence of positive results in our analysis.

3. Features (line 105): While the list of features is provided in Table S1, it is unclear what

these features correspond to based on thei

---

## [Decision Letter · Decision Letter 1]

23 Sep 2024

A Machine-Learning model for prediction of Acinetobacter baumannii hospital acquired infection

PONE-D-24-18204R1

Dear Dr. Ido Neuman,

We’re pleased to inform you that your manuscript has been judged scientifically suitable for publication and will be formally accepted for publication once it meets all outstanding technical requirements.

Kind regards,

*
**Ali Amanati**
*

**Academic Editor**

*
**PLOS ONE**
*

*
**Additional Editor Comments (optional):**
*

I read the revised manuscript ‎

I have no further comments to add. I thank the authors for their detailed ‎‎replies ‎to the reviewers' comments.‎

*
**Reviewers' comments:**
*

Reviewer's Responses to Questions

**Comments to the Author**

1. If the authors have adequately addressed your comments raised in a previous round of review and you feel that this manuscript is now acceptable for publication, you may indicate that here to bypass the “Comments to the Author” section, enter your conflict of interest statement in the “Confidential to Editor” section, and submit your "Accept" recommendation.

Reviewer #2: (No Response)

Reviewer #3: All comments have been addressed

2. Is the manuscript technically sound, and do the data support the conclusions?

Reviewer #2: Yes

Reviewer #3: Yes

3. Has the statistical analysis been performed appropriately and rigorously? 

Reviewer #2: Yes

Reviewer #3: Yes

4. Have the authors made all data underlying the findings in their manuscript fully available?

Reviewer #2: Yes

Reviewer #3: No

5. Is the manuscript presented in an intelligible fashion and written in standard English?

Reviewer #2: Yes

Reviewer #3: Yes

6. Review Comments to the Author

Reviewer #2: The authors have answered my concerns well. In my opinion, this article deserves to be accepted now.

Reviewer #3: (No Response)

7. PLOS authors have the option to publish the peer review history of their article (what does this mean?). If published, this will include your full peer review and any attached files.

Reviewer #2: No

Reviewer #3: No

---

## [Editor Report · Acceptance letter]

27 Sep 2024

PONE-D-24-18204R1 

PLOS ONE

Dear Dr. Neuman, 

I'm pleased to inform you that your manuscript has been deemed suitable for publication in PLOS ONE. Congratulations! Your manuscript is now being handed over to our production team.

Kind regards, 

on behalf of

Professor Ali Amanati 

Academic Editor

PLOS ONE